# Total synthesis of teixobactin

Kang Jin[1], Iek Hou Sam[1], Kathy Hiu Laam Po[2], Du'an Lin[1], Ebrahim H. Ghazvini Zadeh[3], Sheng Chen[2], Yu Yuan[3] & Xuechen Li[1]

To cope with the global bacterial multidrug resistance, scientific communities have devoted significant efforts to develop novel antibiotics, particularly those with new modes of actions. Teixobactin, recently isolated from uncultured bacteria, is considered as a promising first-in-class drug candidate for clinical development. Herein, we report its total synthesis by a highly convergent Ser ligation approach and this strategy allows us to prepare several analogues of the natural product.

[1] Department of Chemistry, State Key Lab of Synthetic Chemistry, The University of Hong Kong, Pokfulam Road, Hong Kong, China. [2] Department of Applied Biology and Chemical Technology, The Hong Kong Polytechnic University, Hung Hom, Kowloon, Hong Kong, China. [3] Department of Chemistry, University of Central Florida, 4111 Libra Drive, Orlando, Florida 32816, USA. Correspondence and requests for materials should be addressed to S.C. (email: Sheng.chen@polyu.edu.hk) or to Y.Y. (email: yu.yuan@ucf.edu) or to X.L. (email: xuechenl@hku.hk).

The globally ever-increasing antimicrobial resistance has become a serious threat to human health and it demands immediate attention to develop novel therapeutic agents[1–3]. Recently, a breakthrough cyclic depsipeptide antibiotic, teixobactin, has been discovered through screening of uncultured bacteria[4]. Teixobactin exhibits excellent activities against an array of Gram-positive pathogens, including methicillin-resistant *Staphylococcus aureus*, vancomycin-resistant *Enterococcus* and *Mycobacterium tuberculosis*. Previous studies have shown that teixobactin binds to a highly conserved motif of lipid II (precursor of peptidoglycan) and lipid III, which is a precursor of the cell wall component teichoic acid, and therefore inhibits bacterial cell wall biosynthesis. Since the molecular target of teixobactin is not an endogenous protein, mutant strains of *Staphylococcus aureus* or *Mycobacterium tuberculosis* that are resistant to teixobactin have not been obtained even after 27 days of repetitive treatments. Compared with other lipid II inhibitors, including vancomycin, ramoplanin, enduracidin and mannopeptimycin (Fig. 1a), teixobactin has a comparatively less complex bioactive pharmacophore structure (Fig. 1b). It remains to investigate how teixobactin binds to lipid II differently from others. Nevertheless, teixobactin is a very promising candidate for the development of new antibacterial drugs and provides a new cyclic depsipeptide-based structural scaffold for medicinal chemistry studies[5–7].

The attractive antibacterial profile of teixobactin in conjunction with its novel structure features has enticed significant interests for its synthetic studies[8–11]. Teixobactin contains a 13-membered depsicyclic ring motif composed of four amino acids including a non-proteinogenic amino-acid residue, enduracididine (Fig. 1), which is anchored with a seven-amino-acid linear chain. Herein, we report the total synthesis of teixobactin via convergent construction.

## Results

**Synthesis of teixobactin.** Teixobactin contains a (2S, 4S) enduracididine (L-*allo*-End), an *N*-methyl-D-phenylalanine and three other D-amino-acid residues. Except L-*allo*-End, all other amino-acid derivatives are commercially available and the analogues of teixobactin containing single-amino-acid substitution (for example, by replacing L-*allo*-End with Arg) have been prepared previously by solid-phase peptide synthesis[8,9]. It goes without saying that the knowledge accumulated during the analogue synthesis and biological evaluation would shed light on the natural product; however, in reality, the fundamental chemical reactivity, structural features and detailed biological function of teixobactin have to be rigorously established based on its natural form, which is the mainstay for any future medicinal chemistry optimization. For this reason, we have recently developed an efficient synthesis of enduracididine building block[10], and have now incorporated it into the total synthesis of teixobactin. To construct teixobactin, we adopted a convergent strategy via Ser/Thr ligation[12] by merger of the linear hexapeptide (1–6) and the cyclic depsi-pentapeptide (7–11). Such a strategy is highly advantageous for the divergent synthesis of teixobactin analogues and allows expedient structure–activity relationship studies of the pharmacophore.

We selected to perform the peptide cyclization at the least sterically congested Thr8–Ala9 site, thus the synthesis of cyclic peptide commenced from Thr8 (Fig. 1b). Observing that Fmoc-Ile-OH was not readily coupled with the resin-bound Alloc-D-Thr to form the ester linkage, depsipeptide Alloc-D-Thr-O(Fmoc-Ile)-OH 4 was first prepared via solution-phase coupling and then immobilized onto the resin (Fig. 2). Alloc and Fmoc were chosen as the orthogonal protecting groups. Alloc-D-Thr-OH was first protected as 4-methoxybenzyl (PMB) ester 2, which was next

coupled with Fmoc-Ile-OH to afford depsipeptide 3. After the PMB group of 3 was removed by trifluoroacetic acid (TFA), the resulting free carboxylic acid 4 was immobilized onto 2-Cl-Trt resin to form 5. On removal of the Alloc group effected by Pd(PPh₃)₄/PhSiH₃, the free amino group was coupled with Boc-Ser(OtBu)-OH under standard conditions to give resin-linked depsi-tripeptide 6. It is noted that during these processes, *O*- to *N*-acyl (Ile) transfer was not observed. Subsequently, the coupling of Fmoc-End(Cbz)₂-OH onto resin-linked 6 with DIC/HOBt was very sluggish, requiring three repetitive 10-h couplings for completion. Next, Fmoc-Ala-OH was elongated onto the resin to afford 7, which was then cleaved from the resin under the mild acidic conditions (TFE/AcOH/DCM) to give the side chain protected peptide 8 for cyclization. Even though the formation of diketopiperazine during the processes of Fmoc removal of the End residue and the subsequent Fmoc-Ala coupling was a major concern for the linear peptide synthesis, we were delight to find out that this side reaction was not observed, probably prohibited by the bulky group. Tetrapeptides (12-member ring) have been known to be difficult for cyclization due to the constrained geometry of the peptide backbone[13]. In the case of teixobactin, the cyclization of the 13-membered depsicyclic ring proceeded smoothly using HATU/HOAt/OxymaPure[9] in CH₂Cl₂ at a concentration of 0.1 mM for 24 h at room temperature. The resultant cyclic peptide was subjected to TFA treatment and hydrogenation (Pd(OH)₂, H₂) to remove the Boc group and Cbz group, respectively. After high-performance liquid chromatography (HPLC) purification, cyclic peptide 9 was obtained in 17% based on the weight of the crude linear peptide 8 obtained from SPPS.

**Study on the epimerization of the cyclization.** To address the potential epimerization problem during the cyclization, we prepared two model acyclic pentapeptides (in both cases, L-*allo*-End was replaced by L-Orn) and investigated the product stereochemistry after ring closure (Fig. 3). In the first model compound, D-Thr (2R, 3S) 1a was used to construct the cyclization precursor, whereas in the second case, L-*allo*-Thr (2S, 3S) 1b was employed to build the quasi-cycle. Both compounds underwent smooth cyclization under the previously described conditions, and the epimeric products gave distinctive retention times when co-injected in the HPLC analysis (Fig. 3). This observation supported that the stereochemical integrity was retained during the preparation of depsipeptide 9.

Having completed cyclic peptide part 9, Ser ligation was then used to complete the synthesis of teixobactin. The linear peptide fragment containing C-terminal salicylaldehyde ester 11 was synthesized by solid-phase peptide synthesis[13]. Ligation proceeded efficiently to couple 9 and 11 in pyridine/AcOH (6:1, mol:mol) with a completed HPLC conversion to afford teixobactin 12 in 37% yield after HPLC purification (Supplementary Methods). The general applicability and robustness of this strategy was further demonstrated by the synthesis of several analogues 13–16. By replacing L-*allo*-End, D-Thr and D-*allo*-Ile, analogues 13 (D-*allo*-Ile5D-Ile), 14 (D-*allo*-Ile5D-Ile, End10Arg), 15 (D-*allo*-Ile5D-Ile, D-Thr8Thr, End10Arg) and 16 (End10Orn) were prepared accordingly. It was noted that the solid-phase peptide synthesis of the End10Arg and End10Orn analogues was much smoother than that of teixobactin, leading to 30% and 50% yields (based on the weight of the crude peptide obtained from SPPS), respectively, after cyclization. This observation suggests that the synthesis of teixobactin in its natural form is likely to come across a distinct set of challenges compared with the analogue synthesis; and the natural product should be the rational starting point for future chemical and biological optimization.

**Figure 1 | The structures of some cyclic peptide-based lipid II inhibitors. (a)** Enduracidins A and B were discovered in late 1960s by Nakazawa and Mizuno et al[14,15]. The structure of Ramoplanin was established in 1989 by Cavalleri et al[16]. Mannopeptimycin was isolated in 1950s and first elucidated around 2002 (ref. 17). **(b)** Teixobactin was discovered in 2015 as a new lipid II inhibitor[4]. Our synthetic disconnection of teixobactin was also described.

**Antibacterial studies.** Synthetic teixobactin and its analogues were subjected to the assessment of their antimicrobial activities by determining their minimal inhibitory concentration on various Gram-positive bacteria strains including methicillin-susceptible *S. aureus* strain ATCC29213 and methicillin-resistant *S. aureus* clinical isolates. Our data indicated that the synthetic teixobactin exhibited similar activity as the natural teixobactin, as reported previously (Table 1) and the analogues with different stereo-configuration at D-*allo*-Ile exhibited reduced activity towards different Gram-positive bacterial pathogens. L-*allo*-End is important for the potent antibacterial activity of teixobactin, as its replacement with Orn (**16**) or Arg[14] caused reduced antibacterial activities. These studies suggested that superior teixobactin analogues should maintain the L-*allo*-End-cyclic peptide fragment and examine the variation of the seven-amino-acid linear chain fragment. In this effort, the original stereo-configuration of the amino-acid residue should be thoroughly scrutinized. From our study, it is encouraging to find that Orn-containing teixobactin analogues maintained modest antibacterial activity. Since Orn is readily available as compared with L-*allo*-End and the synthesis of the Orn-containing cyclic depsipeptide (**7–11**) gave the best yield among all the analogue syntheses, which is amenable to a large-scale synthesis. Through the above-developed convergent synthetic strategy, various linear hexapeptide (**1–6**), such as **11**, can be readily prepared and ligated to the Orn-containing the cyclic depsi-pentapeptide (**7–11**) to investigate the contribution of each amino acid in the linear peptide fragment towards the antibacterial activity, which will lead to the optimal structure for the development of teixobactin with improved properties.

## Discussion

In summary, we have developed the total synthesis of teoxobactin featuring a ligation-mediated convergent 6 + 5 strategy, which aims to provide a combinatory approach for the synthesis of its analogues to establish a comprehensive structure–activity relationship and facilitate future search for analogues with improved pharmacological properties. Our preliminary structure-activity relationship (SAR) studies suggested that replacing of difficultly obtained End with readily available Orn could maintains a modest antibacterial activity, which provides alternative structural motif for its medicinal chemistry studies.

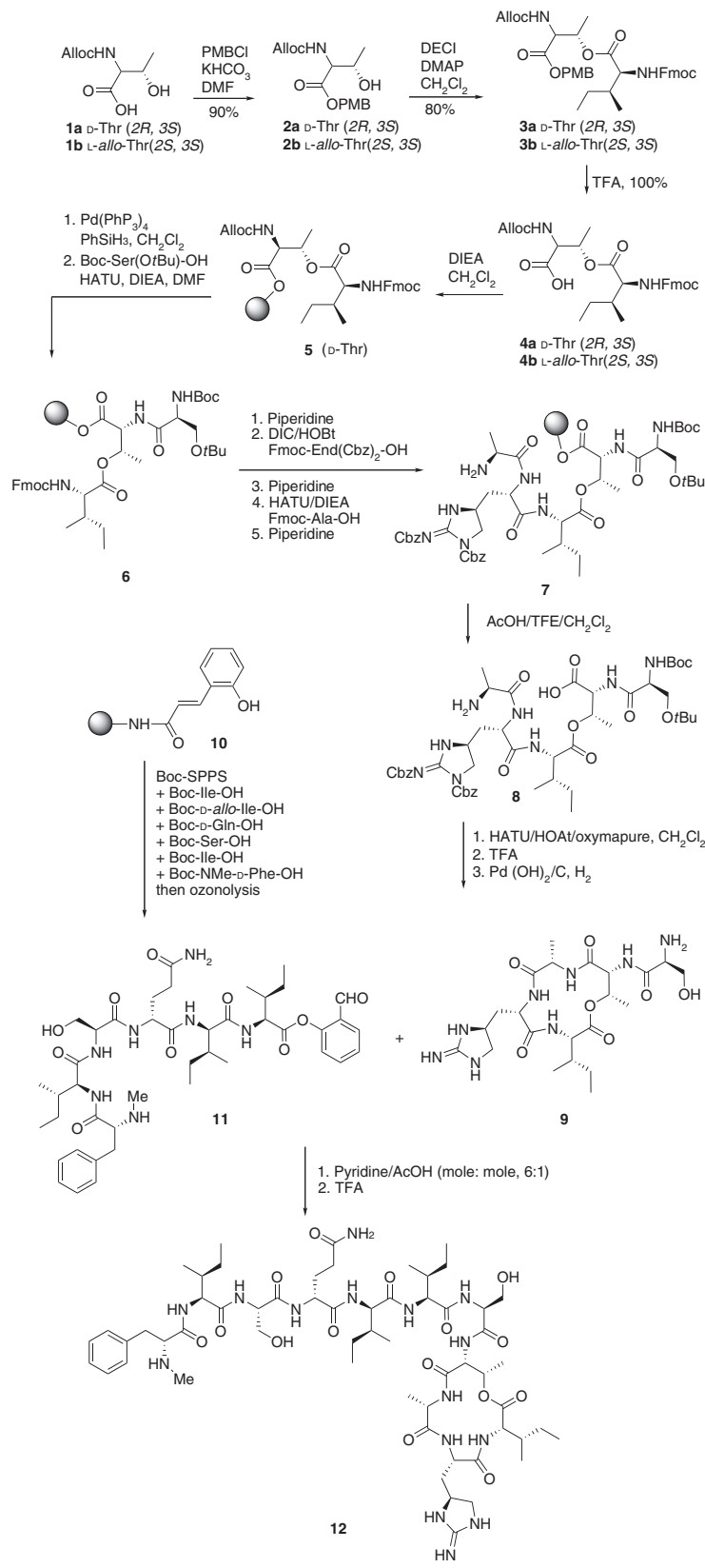

**Figure 2 | Synthetic route towards teixobactin.** Peptides 9 and 11 were prepared by SPPS and coupled through Ser ligation, which was developed in our group.

## Methods

**General.** For nuclear magnetic resonance (NMR) data and mass spectra of the compounds in this article, see Supplementary Figs 1–43, and Supplementary Tables 1 and 2. High-resolution mass spectra were recorded on a quadrupole-time-of-flight mass spectrometer. Accurate masses are reported for the molecular ion $[M + H]^+$ or a suitable fragment ion. NMR spectra were recorded at room temperature on a 400-MHz NMR spectrometer or Bruker Avance 600 FT-NMR spectrometer at 600 MHz.

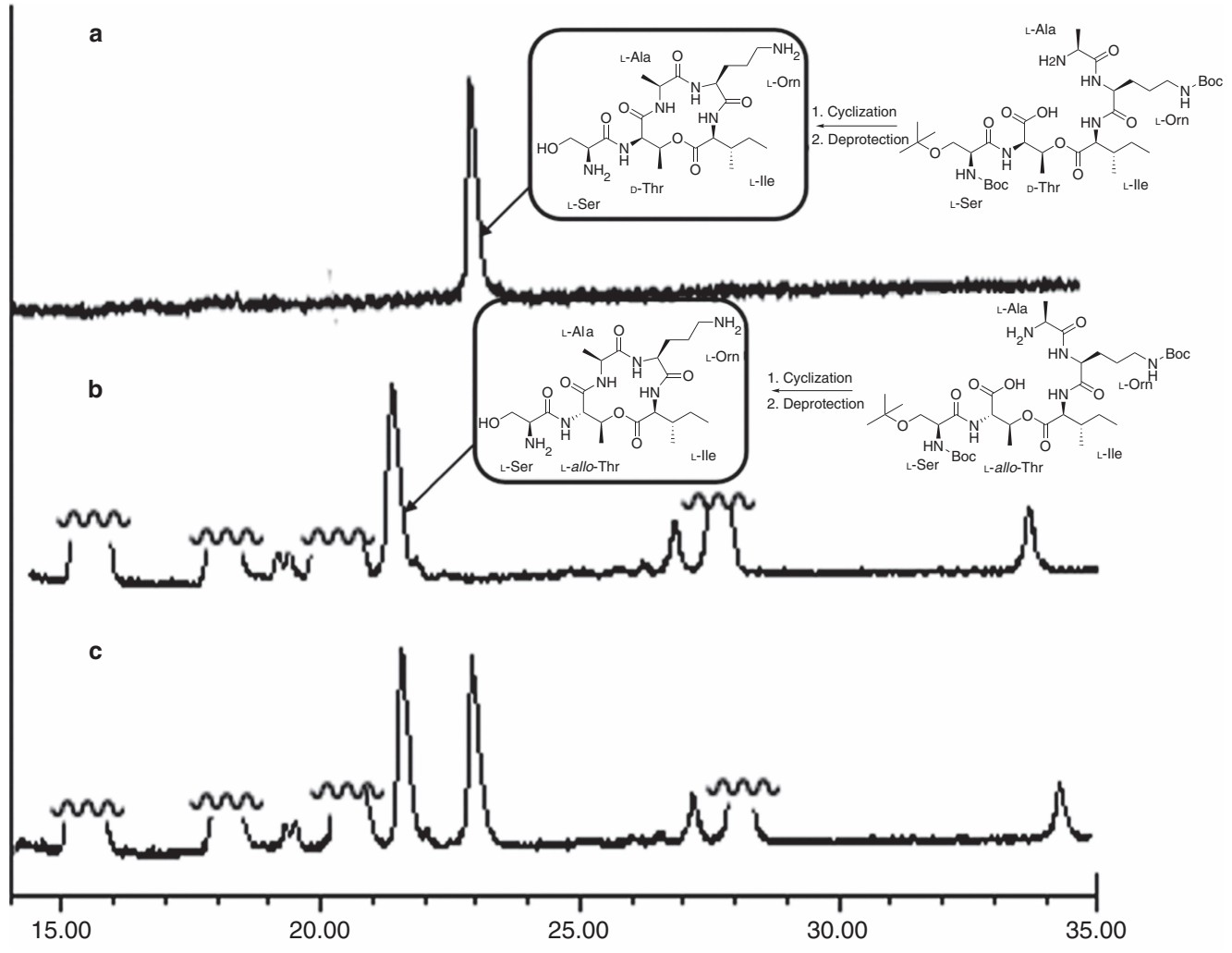

**Figure 3 | HPLC spectra of peptide cyclizations.** (**a**) Purified cyclic peptide with D-Thr. (**b**) Crude cyclization mixture of the one with L-*allo*-Thr. (**c**) Co-elution of **a** and **b**.

**Table 1 | Antibacterial activities of the synthetic teixobactin and its analogues.**

| Bacterial strains | MIC (µg ml$^{-1}$) | | | | | |
|---|---|---|---|---|---|---|
| | **12** | **13** | **14** | **15** | **16** | **teixobactin** |
| *S. aureus* ATCC29213 | 0.5 | 4 | 16 | >32 | 2 | 0.5 |
| MRSA | 0.5 | 4 | 8 | >32 | 2 | 0.5 |

MIC, minimal inhibitory concentration.

**Convergent synthesis of teixobactin and its analogues via serine/threonine ligation.** The peptide salicylaldehyde ester (1.5 equiv.) and the cyclic peptide fragment (1.0 equiv.) were dissolved in pyridine/AcOH buffer (6:1, mol:mol). The reaction was monitored by HPLC mass spectrometer until the reaction was completed, followed by acidic cleavage of *N,O*-benzylidene acetal intermediate by TFA cocktail (TFA/H$_2$O/TIPS, 95:2.5:2.5, v:v:v). The residue was purified by preparative HPLC.

**Data availability.** The authors declare that the data supporting the findings of this study are available within the article (and its Supplementary Information files).

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

## Acknowledgements

This work was supported by the Research Grants Council-Collaborative Research Fund of Hong Kong (C7038-15G) and the University of Central Florida. We thank Mr Edward Ramos for the preparation of some enduracididine intermediates. We thank Professor Kim Lewis from Northeastern University for providing an authentic teixobactin sample.

## Author contributions

X.L., K.J. I.H.S and D.L. designed the synthetic route and implemented the chemical synthesis. Y.Y and E.H.G.Z implemented the End building block synthesis, S.C. and K.H.L.P. performed the biological studies. X.L. conceived of the idea and wrote the paper.

## Additional information

Competing financial interests: The authors have filed a provisional patent application related to this work.

