## [Peer review file · Nature Communications]

Reviewers' comments:

Reviewer #1 (Remarks to the Author):

The manuscript of Li and co-workers describes the first total synthesis of Teixobactin. Authors used a convergent approach based in the used of two unoriented fragments through the Ser ligation technique that has been developed earlier by the same authors. The use of this methodology will open the possibility of synthesizing a large number of analogues in a short period of time. The study regarding the tentative racemization during the cyclization is really well done and appreciated by this referee. The present manuscript deserves publication in Nature Communication.

Queries:

What means the low efficiency of on-resin ester formation, depsipeptide Alloc-D-Thr-O(Fmoc-Ile)-OH was first prepared via solution phase coupling?

Please explain the rationale of the analogues synthesized? Many changes in just one analogue?

Any DKP formation during the removal of the Fmoc of the End residue and the incorporation of Fmoc-Ala?

Reviewer #2 (Remarks to the Author):

This paper describes the full synthesis of teixobactin with reasonable yield, an important advance that will help in therapeutic development of this interesting novel class of antimicrobials. The authors also synthesize a limited number of analogs with considerable retention of activity, a good indication for the feasibility of modifying the parent compound.

The writing is generally acceptable, but the manuscript does need editing. For example, "...all other amino acid derivatives are commercially available and the analogues of teixobactin contains single amino acid mutation..." - plural and singular cases are mixed up, and "mutation" is inapplicable here, should be "substitution".

The authors report about a 3 fold lower activity of synthetic teixobactin as compared to the published compound. Teixobactin adheres to hydrophobic surfaces, and the MIC reported in Ling et al. was determined in the presence of polysorbate to mitigate this problem. The authors should repeat the assay in the presence of polysorbate.

Reviewer #3 (Remarks to the Author):

The authors have demonstrated the first total synthesis of teixobactin in this paper. They prepared ester 3 in advance to the solid-phase peptide synthesis of linear peptide 8. During the synthesis of 8, they found that three repetitive coupling with an endoracididine unit that is a rare amino acid was required. After successful macrolactamization of cyclotetradepsipeptide 9, the Ser moiety was coupled with a hexapeptide side chain by way of the salicylaldehyde mediated coupling they originally developed. After deprotection, the total synthesis of teixobactin was achieved.

The authors were worried about the epimerization at the C-terminus during macrolactamization of the cyclotetradepsipeptide, thereby, they utilized a model system to prove the macrolactamization was performed without epimerization at the D-Thr moiety.

They also synthesized analogues of teixobactin according to this convergent method and evaluated their antibacterial activities. They found that allo-endoracididine can be replaced by Orn and D-allo-Ile is significant to retain the biological activity.

Because teixobactin did not give any mutant that has resistant to teixobactin, it is an attractive and novel antibiotic. The paper describing the total synthesis as well as those of analogues and their SAR study could have a notable impact on readers. Publication is recommended as it stands.

A detailed point-by-point response to the reviewers' comments is as follows.

Reviewer 1:

1. What means the low efficiency of on-resin ester formation, depsipeptide Alloc-D-Thr-O(Fmoc-Ile)-OH was first prepared via solution phase coupling?

Response: *it has been re-written as " Observing that Fmoc-Ile-OH was not readily coupled with the resin-bound Alloc-D-Thr to form the ester linkage, depsipeptide Alloc-D-Thr-O(Fmoc-Ile)-OH 4 was first prepared via solution phase coupling and then immobilized onto the resin."*

2. Please explain the rational of the analogues synthesized? Many changes in just one analogue?

Response: *At this stage, the synthesis of the analogue with multiple changes is to test the tolerance and robustness, general applicability of the synthetic strategy. "The general applicability and robustness of this strategy was further demonstrated by the synthesis of several analogues" is added into the text.*

3. Any DKP formation during the removal of the Fmoc of the End residue and the incorporation of Fmoc-Ala?

Response: *it is added "It is noted that during the processes of the removal of the Fmoc of the End residue and the incorporation of Fmoc-Ala, the formation of diketopiperazine was not observed."*

Reviewer 2:

4. "...all other amino acid derivatives are commercially available and the analogues of teixobactin contains single amino acid mutation..." - plural and singular cases are mixed up, and "mutation" is inapplicable here, should be "substitution".

Response: corrected

5. The authors report about a 3 fold lower activity of synthetic teixobactin as compared to the published compound. Teixobactin adheres to hydrophobic surfaces, and the MIC reported in Ling et al. was determined in the presence of polysorbate to mitigate this problem. The authors should repeat the assay in the presence of polysorbate.

Response: we redid the MIC with the suggested condition. The synthetic teixobactin has identical antibacterial activity as the authentic teixobactin.

REVIEWERS' COMMENTS:

Reviewer #1 (Remarks to the Author):

The author made all the changes requested by the reviewer

Reviewer #2 (Remarks to the Author):

The authors have adequately responded to critiques.